# A Health Support Model for Suburban Hills Citizens

Telung Pan

Bachelor Program in Interdisciplinary Studies, College of Future, National Yunlin University of Science & Technology, Yunlin 64002, Taiwan; telung@yuntech.edu.tw

**Abstract:** In spite of the increasing understanding of the importance of social support attached to health and medical services in suburban areas, a support model needs to be established that can benefit those areas with different living patterns. To that end, this research employs both quantitative and qualitative analyses to investigate the medication, social support, and care forms and sources in the suburban hills of an area in a central Taiwan village. Different types of data sources were collected during the analysis phase to develop a support model. A new model integrated with the boundaries between the demander and the provider was developed. An experimental mobile app was also developed, and this app was based on the concept of this new support model. It is hoped that the medicine service, care support, and key information on society health activities could be provided by means of using the easiest and simplest GUI through which the local healthcare economic system could be further established.

**Keywords:** health support model; suburban hills; society health care





## 1. Introduction

In Taiwan, the National Health Insurance Program has been implemented for more than 20 years. Under this program, the health insurance coverage rate has reached approximately 99% of the population [1]. However, the medical services or infrastructures provided in some village areas or suburban hills still remain insufficient. The major health issues in rural areas are mainly associated with the problems of chronic shortages of doctors, dentists, pharmacists, and a widening gap in life expectancy. In this study, when compared with those ordinary people residing in the urban areas, their counterpart suburban hill citizens had been afflicted with these aforesaid problems [2].

This article describes the situation in a suburban hill area in Taiwan, and it is concerned with the comprehensive medical care for local villagers and local residents. That is because they require regular assistance from medical specialists in their homes in order for them to maintain their health and manage their health problems. This does not include the hospital-at-home care provided by a professional on a hospital team.

The meanings, the material conditions, and the social relations of both the domestic life and the healthcare work in relation to those care services ranging from hospital medical care to home health care are changing [3]. Little is known about the total care received by those residents living in suburban hills. This is not only including what happens to them when they are at home but also the care they receive at home. What is more, there is a lack of knowledge about what factors would mainly influence the care given in these areas.

Scholars in this field cannot agree on a universally acceptable definition of the construct of how to access healthcare [4]. This is attributed from an insufficient understanding of the roles played by different facilitators and the barriers between urban and the rural specialty care [5]. The elderly, women, children, racially and socioeconomically disadvantaged groups, and individuals with chronic health conditions have faced special challenges in their disproportionate access to specialty care and poorer health outcomes despite the impact of geographic residency status, especially in medically underserved urban and rural areas [6].

Although there are many definitions of urban and rural geographic areas, the construct of the suburban hills has been regarded as an important concept after the Satoyama Initiative was firstly introduced in the year 2015 [7]. This is a global initiative based on the concept of Satoyama, referred to as a traditional rural landscape in Japan. The initiative promotes the concept as such and helps integrate conservation and sustainable use of biodiversity in production landscapes, outside of the protected areas. The initiative has been supported and implemented by an international partnership comprising over 100 governments, civil society organizations, and indigenous peoples [7].

Regardless of the vast differences between the urban and rural landscapes, several characteristics are shared among vulnerable populations. One of the important concepts is "approachability". Healthcare approachability represents the capacity of a health system designated to identify and provide services in need, such as transparency, information, and screening [8]. Another important concept associated with suburban hill medical services is "availability and accommodation". This concept relates to the timely attainment, geographic locations, hours of operation, and capacity of services offered. Traveling for care is also deemed as a key issue because traveling to larger cities to receive care is a burdensome task for rural residences [7,8].

Accordingly, the study aimed to exam practices and concerns related to medical issues in a suburban hill area. Given the exploratory nature of this study and its natural limitation in the area, a mixed-method approach combined with both qualitative and quantitative thinking was employed to establish the model proposed in this research. Specifically, we aimed to obtain a comprehensive understanding of different experiences with and concerns about medical needs. To that end, qualitative data would provide a clear explanation and elaborate accounts of quantitative data. Information derived from this study will help inform healthcare administrators and policymakers of those practices related to and concerns about medical services in rural areas.

## 2. Materials and Methods

This study was carried out in a suburban hill area of central Taiwan with 1202 inhabitants, and most of them were 65 years of age or older. Table 1 shows basic statistics of those people in the village. No medical care facilities were provided in that village. During the time when this study was conducted, people seeking medical assistance had to spend for more than 2 h on bus transportation to the nearest town or city.

**Table 1.** Demography data of population in Guilin village.

| Gender | Persons | % |
|---|---|---|
| Male | 657 | 54.84% |
| Female | 545 | 45.16% [1] |

[1] Data source: https://dounan.household.yunlin.gov.tw.

### 2.1. Model Developed Based on the Facts: Quantitative and Qualitative Data Analyses

The aim of this study was to develop a model in order to investigate the community's health living conditions in suburban hills. The Statistics obtained from official data, online surveys, review of literature, focus groups, and field studies were used to identify and describe the health and medical demands for the villagers. After informed consent was obtained, the participants completed an Internet survey developed by the researchers to assess the demographics and variables of living, including health status, diseases, financial conditions, entertainment, outdoor activities and frequency, and questions such as "feel lonely?", "feel happy?", and "friend to talk?" etc. The survey was available from 1 April to 1 May 2020 with continues reminders given to all participants. A sample of 77 villagers responded, and all participant data were entered into an excel file and analyzed, using SPSS software and python programming for data visualization.

*2.2. Field Study and Focus Groups*

In order to further clarify and elaborate upon the quantitative data, this research involved three field studies in the suburban hills along with one focus group discussion conduced with seven participants. The potential participants were recruited from a local villager permanently residing in the field and local health service providers, e.g., pharmacists, social workers, and care givers. The researchers explained the goal of the study to the group of participants before they were engaged in a semi-structured, audio recorded discussion, and photos were taken for nearly two hours. Each of the five individual researchers assigned meaningful codes and identified concepts embedded in the collected data. Afterwards, they collectively resolved debatable codes [9]. The concepts identified as the main factors to form the follow-up suburban hills health support model are provided below. Table 2 shows the examples of meaningful codes in relation to the conceptualization in qualitative research.

**Table 2.** Examples of meaningful codes for the main concepts in qualitative research.

| No | Key Words | Original Text | Concepts | Weighting |
|----|-----------|---------------|----------|-----------|
| 1 | dialysis | "Someone is always doing dialysis nearby." | Medical needs | 3 |
| 2 | sports | "Usually a group of people listen to the radio or take exercise at the courtyard in front of a temple." | Society support | 3 |
| 3 | take care of grandchildren | "Because my son works in another city, I need to help take care of my own grandson." | Family burden | 1 |
| 4 | clinics | "The nearest clinic for me to go to is located in the downtown area of Gukeng" | Medical needs | 3 |
| 5 | free tuition courses | "The nearby school offers free tuition courses and dinner." | Society support | 3 |
| 6 | grocery shopping | "I usually take the bus to Douliu city if I have to buy food. Alternatively, I eat homegrown vegetables sometimes." | Traffic demand | 2 |
| 7 | chatting | "I usually have a chat with neighbors in other people's places." | Society support | 3 |
| 8 | hospital | "If you want to a larger hospital, you have to go to Douliu City." | Medical needs | 3 |

In the previous step, we presented a picture of both the citizens living in suburban hills and the primary health demand of those villagers. Most of the villagers are elders, 20.71% are older than 65. Many of them had issues with life functions, e.g., dialysis needs, medication supplies, intergenerational education, need social support, and transportation. Figure 1 represents the data collection and analysis processes for this research.

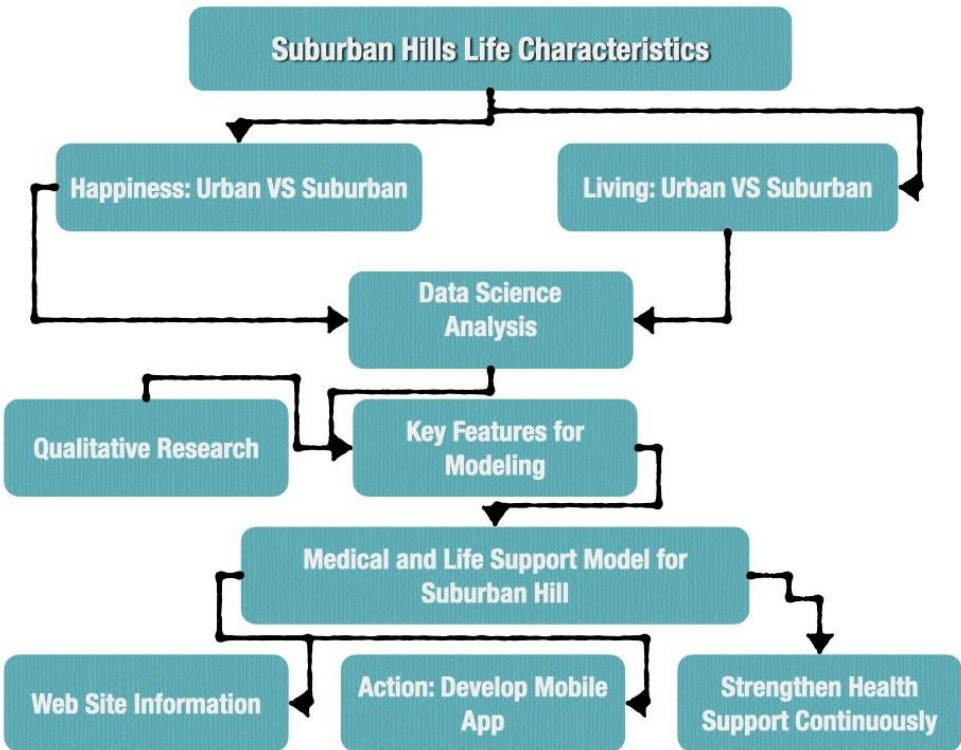

**Figure 1.** Data collection and analysis process.

### 3. Results

The data obtained from the online survey and the qualitative data were collected from field studies and a focus group. The pharmacists and social workers in the focus group provided expert opinions and thus helped establish a demand and supply model. The new model was found to provide a solution for the needs of health and life support in the community. The old pharmacists or retired volunteers in the community did not know where the patients in need were. That was because the space of the suburban hills is vast, and the living quarters of residents are scattered. On the other hand, the elderly people in need of medical care did not know how to seek medical assistance from those experts available to help.

The survey results showed that the elderly people had their own approaches to make themselves live healthier. The most principal factor for them to live healthy was trying to "avoid loneliness," which means to have more opportunities to commune with neighbors, friends, or relatives. The key factors were exactly the same with those identified by means of the opinions given by the expert focus group, and some of them had the experiences in providing older people with care either at organizations or at homes for over 20 years.

To combine the results derived from the quantitative analysis with those of the qualitative analysis, a model for the health life support in suburban hills was established and based on key factors. Figure 2 shows the key factors considered best to support this new model. The top three factors included medical care, transportation, and society.

We developed a different repertoire of strategies and methods based on the new model to face these issues. According to the needs of those residents living in the suburban hills, this model was established and based on their demands and those suppliers needed to provide better health medical services.

In order to verify whether this model was workable or not, we started with the establishment of the continuous prescription for chronic diseases, materials, and health activities. This was made with the cooperation of the local pharmacists, social workers, and retired volunteers. After several visits in advance and the establishment of multiple roles through consensus, a useful and easy-to-use mobile app was developed, using Google

Flutter framework based on Dart programming language, which can function for both Android and iOS apps. In this study, the main feature of the program solely ran on a large and easy-to-use interface. When residents of the suburban hills had medical needs, they could press an easy-to-use button, and the corresponding pharmacists or social workers would contact the residents in need after receiving the message sent from the app installed on their mobile phones. Figure 3 illustrates the main interface and the functions of this designed mobile app.

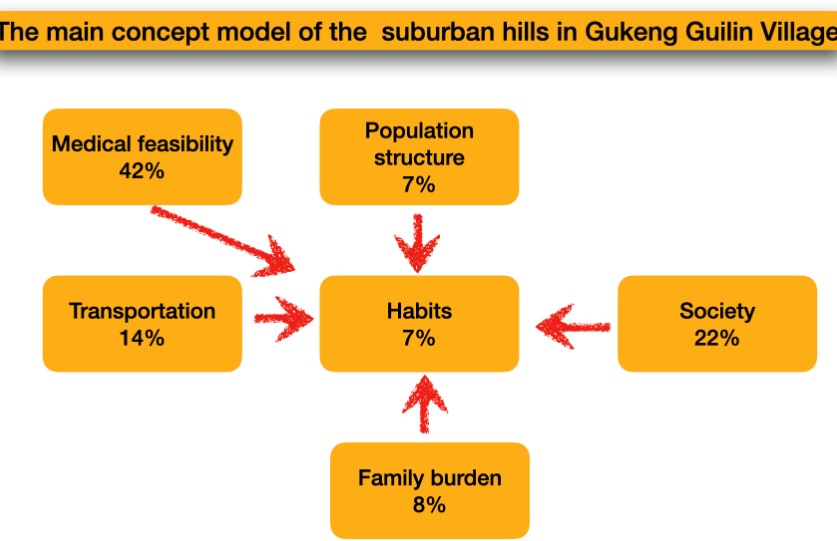

**Figure 2.** Key factors supporting the new model.

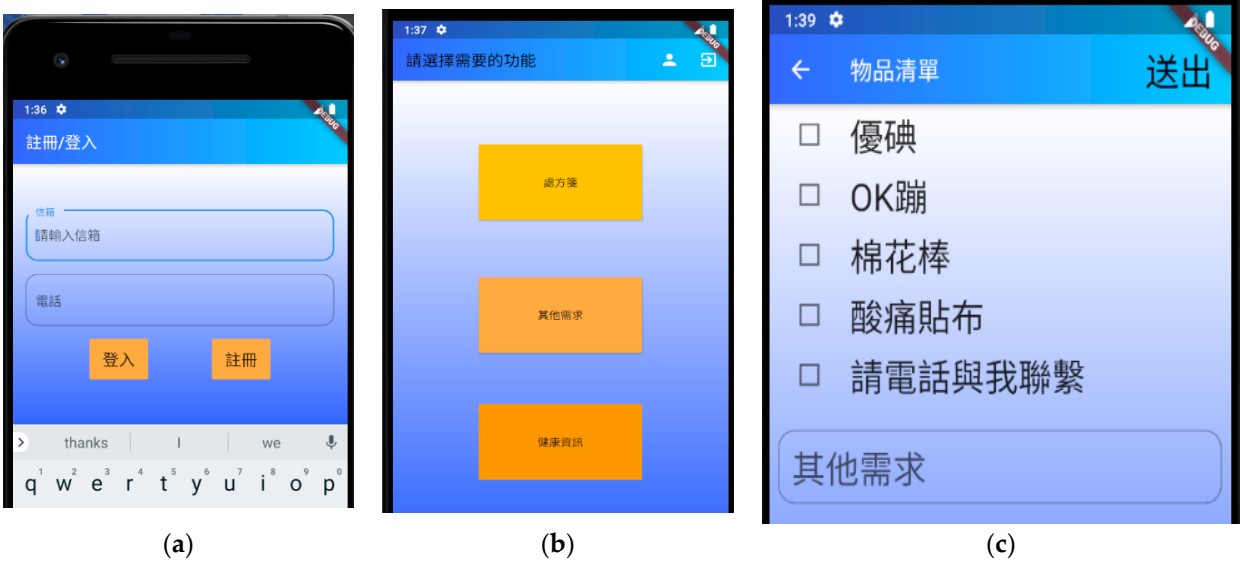

**Figure 3.** Main interface and functions design in traditional Chinese character. (**a**) is registration and login in screen by key in phone number; (**b**) is function selection, three large buttons provide medicine, daily necessities, and health consultation respectively; (**c**) provide items selection without input words.

In analyzing the results, three knowledge categories appeared: (a) suburban hills health care can be helped by local revitalization, (b) integrate existing local resources, and (c) made beneficial use of modern technology. Each category was unique in terms of the problems it addressed and the source from which it came.

### 3.1. Suburban Hills Health Care Helped by Local Revitalization

In the US, nonprofit hospitals are encouraged to collaborate with local public health experts in the conduct of community health needs assessments (CHNAs) for the larger goal of improving community health [10]. In research in India, traditional medicine (particularly herbal medicine) is considered as a major healthcare provider around the globe, notably in rural and remote areas. Indian traditional medicinal systems like Ayurveda, Siddha, and Unani have an extraordinarily rich history of effectiveness. Such medicine is important for the people in India, and it is important to promote such medicine and to integrate them into clinical practice [11]. Yet, little is known about whether collaborations between local health departments and hospitals may be beneficial to community health, as most of the research is focused on other topics, e.g., economy, tourism, and agriculture [12–18]. In this study, we developed a similar model and implemented a pilot mobile app to provide evidence for the model.

Stronger collaboration between nonprofit hospitals and suburban hills citizens was positively associated with healthier individual-level behaviors. Social capital may also play a moderating role in improving individual and population health.

### 3.2. Integrate Existing Local Resources

Social determinants of health are the major drivers of health and disparate health outcomes across communities and populations. Given this, this study asserts that integrating existing local resources in a suburban hill, for example retired medical physicians, pharmacists, and social workers, should become a vital part in the establishment of health care. Although the most effective approaches to integrate resources in the area need investment of human efforts in advance, the medical service model can be continuous after establishment. These include universalization of the material, integration into clinical education, identification of technology methods for model operation, and continuous improvement. This research highlights an example of the service of continuous prescription for chronic disease by using a mobile app, thereby connecting the demand for citizens and medical suppliers.

### 3.3. Made Beneficial Use of Modern Technology

Digital health solutions can be grouped into categories: remote access to specialists, building and supporting local ability, and patient-directed interventions. Limited evidence exists for most digital solutions, specifically in suburban hills, although examples and pilot projects have been described.

While the use of digital health solutions for suburban hills is needed, there is a limited evidence base for most of them. Future efforts to expand the use of digital solutions in this population should adhere to best practices for the delivery of health services.

### 4. Discussion and Conclusions

In this paper, we demonstrated an approach to the suburban hills health and medical service model based on evidence from both quantitative and qualitative data. We called this new model the 'Giiobo (Medicine Package)' model. This model can support the residents who in need of the medical support: chronic disease drug supply, supports from local pharmacies, pharmacists, and social workers. Organic place-making emerges through local individual specialists will make the model relaistic This model can re-integrate into a new substantial support ecosystem by the operation of this new design mobile app.

During the development of this project, there were some local residents who were students of a nearby university; they came to help as volunteers, and they have the talents of software development. In the future, hopefully, this Giiobo model can continue to be used by local residents. There are lots of functions that can be integrated and improved in the next version of this mobile app

While digital health solutions are acceptable in the fields of suburban hills, there are also concerns alongside distinct cultural and health beliefs [19]. Youth and innovators

are finding ways to use digital applications and social media, to share ideas, promote positive behavior change, and generate healthy social networks, but these may not be suitable for the elders of suburban areas. Our finding confirms those of statements [20], that using innovative technology to provide healthcare to suburban populations should strive to learn about the historical influences, and current preferences and values, of the local culture. Digital health solutions, like other healthcare services, need to be adapted and implemented with careful consideration of local values and community needs, and more rigorous evaluation is needed.

**Funding:** This research was funded by Center for University Social Responsibility, Ministry of Education, Taiwan, grant number 109-N02-1.

**Institutional Review Board Statement:** Not applicable.

**Informed Consent Statement:** Informed consent was obtained from all subjects involved in the study.

**Data Availability Statement:** Not applicable.

**Conflicts of Interest:** The authors declare no conflict of interest.

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
