# Peer review of "A Health Support Model for Suburban Hills Citizens"

_asi, doi:10.3390/asi4010008_

Round 1
Reviewer 1 Report
The language and the statistics has to be improved.
In order to test if the observed differences are statistically significant some statistical tests have to be applied.
The flow of information exposed needs to be improved.
Author Response
- The language and the statistics has to be improved.
Response: The language has been revised again. - To test if the observed differences are statistically significant some statistical tests must be applied.
Response: Statistics method in this study is using for understanding the characteristics and the differences of suburban hill area and urban area, and is only using at the data collection and analysis before model developing. The quantitative and qualitative data collected are the evidence-based support for the new model, so there is no comparison data before and after the modeling and app implementation. To test if the new model has the differences statistically significant will be conducted in the future study. - The flow of information exposed needs to be improved.
Response: The flow of information exposed have made some revised.
Reviewer 2 Report
The article sent for review seems to be of interest.
The authors should make it clearer whether there is evidence from previous experiences.
Also, since there is a high cultural background in the implementation of this software, it should be clear what problems have arisen and whether the authors consider that it can be of universal or only local application. Likewise, it should be studied which are the modifications that can be developed so that the application can be transmitted to other cultures or health systems.
Author Response
- The article sent for review seems to be of interest.
Response: Thanks a lot for the confirmation by the reviewer. - The authors should make it clearer whether there is evidence from previous experiences.
Also, since there is a high cultural background in the implementation of this software, it should be clear what problems have arisen and whether the authors consider that it can be of universal or only local application.
Response: Related discussion and related content have been added at the discussion and result section, and the evidences from earlier experiences was also added. - Likewise, it should be studied which are the modifications that can be developed so that the application can be transmitted to other cultures or health systems.
Response: What the modifications that can be developed have been added that the application can be transmitted to other cultures or health system.
Round 2
Reviewer 1 Report
Nothing more to add